# MELD-GRAIL-Na Is a Better Predictor of Mortality Than MELD in Korean Patients with Cirrhosis

**DOI:** 10.3390/medicina59030592

**Published:** 2023-03-16

**Authors:** Jung-Woo Kim, Jeong-Han Kim, Won-Hyeok Choe, So-Young Kwon, Byung-Chul Yoo

**Affiliations:** 1Department of Internal Medicine, Konkuk University Medical Center, Seoul 05030, Republic of Korea; 2Research Institute of Medical Science, Konkuk University School of Medicine, Seoul 05030, Republic of Korea

**Keywords:** Child–Pugh score, MELD, mortality, Korean, renal function

## Abstract

*Background and Objectives*: The Child–Pugh (CP) score and Model for End-Stage Liver Disease (MELD) are classical systems for predicting mortality in patients with liver cirrhosis (LC). The MELD-GFR assessment in liver disease–sodium (MELD-GRAIL-Na) was designed to better reflect renal function and, therefore, provide better mortality predictions. This study aimed to compare the prediction accuracy of MELD-GRAIL-Na compared to CP and MELD in predicting short-term (1- and 3-month) mortality in Korean patients. *Materials and Methods*: Medical records of patients with LC admitted to the Konkuk University Hospital from 2015 to 2020 were retrospectively reviewed. Predictive values of the CP, MELD, and MELD-GRAIL-Na for 1-month and 3-month mortality were calculated using the area under the receiver operating curve (AUROC) and were compared using DeLong’s test. *Results*: In total, 1249 patients were enrolled; 102 died within 1 month, and 146 within 3 months. AUROCs of CP, MELD, and MELD-GRAIL-Na were 0.831, 0.847, and 0.857 for 1-month mortality and 0.837, 0.827, and 0.835 for 3-month mortality, respectively, indicating no statistical significance. For patients with CP classes B and C, AUROCs of CP, MELD, and MELD-GRAIL-Na were 0.782, 0.809, and 0.825 for 1-month mortality and 0.775, 0.769, and 0.786 for 3-month mortality, respectively. There was a significant difference between CP and MELD-GRAIL-Na in predicting 1-month mortality (*p* = 0.0428) and between MELD and MELD-GRAIL-Na in predicting 1-month (*p* = 0.0493) and 3-month mortality (*p* = 0.0225). *Conclusions*: Compared to CP and MELD, MELD-GRAIL-Na was found to be a better and more useful system for evaluating short-term (1- and 3-month) mortality in Korean patients with cirrhosis, especially those with advanced cirrhosis (CP class B and C).

## 1. Introduction

The Child–Pugh (CP) and Model for End-Stage Liver Disease (MELD) scores are well-known predictors of wait-list mortality in patients with cirrhosis [1]. Many attempts have been made to modify and improve the MELD system by adding more variables and adjusting the coefficients, leading to the conception of MELD-Na, Refit MELD, and Refit MELD-Na [2,3,4]. Renal function is strongly associated with mortality in patients with cirrhosis. Acute renal failure is relatively common, occurring in approximately 20% of hospitalized patients with cirrhosis and is associated with a 7-fold increase in mortality [5,6]. Therefore, accurately estimating renal function is essential.

Serum creatinine is not an accurate biomarker for estimating kidney function in patients with liver cirrhosis because creatinine synthesis is reduced in cirrhotic patients, and other non-renal variables interfere with creatinine [7,8]. Accordingly, novel ideas to estimate the glomerular filtration rate (GFR) of patients with liver disease have arisen. The model for Glomerular Filtration Rate Assessment in Liver Disease (GRAIL) was introduced in 2018 and includes various variables, including creatinine, blood urea nitrogen, age, sex, race, and albumin, to estimate GFR [9]. Subsequently, in 2020, the Model for End-Stage Liver Disease–Glomerular Filtration Rate Assessment in Liver Disease–Sodium (MELD-GRAIL-Na), which better reflects renal function using GRAIL, was introduced to better predict mortality in patients with cirrhosis on the liver-transplant waiting list [10].

Since GRAIL and MELD-GRAIL-Na were studied among institutions in the United States of America, this study aimed to compare the prediction accuracy of MELD-GRAIL-Na compared to CP and MELD scores in predicting short-term (1- and 3-month) mortality in Korean patients with liver cirrhosis.

## 2. Materials and Methods

All patients diagnosed with liver cirrhosis who were admitted to the Konkuk University Hospital between January 2015 and December 2020 were enrolled and retrospectively reviewed. We excluded patients aged <18 years, diagnosed with hepatocellular carcinoma, other primary or metastatic malignancies, and a primary cause of death unrelated to cirrhosis (i.e., heart failure, coronary artery disease, and pneumonia). Moreover, we excluded patients who did not have medical records within 3 months of initial admission. Liver cirrhosis was defined and diagnosed based on the presence of radiological and ultrasonographic features of cirrhosis including thrombocytopenia, ascites, esophageal/gastric varices, and hepatic encephalopathy. Alcoholic liver cirrhosis was suspected in patients with a history of long-term and continuous alcohol intake of more than 100 g per day (corresponds to 6–7 drinks daily, with each drink containing 13–15 g of alcohol). Chronic hepatitis B was diagnosed as HBsAg being detected positive for more than 6 months. Chronic hepatitis C was diagnosed when an anti-hepatitis C virus (HCV) antibody was present with an HCV-RNA or the recombinant immunoblot assay was positive. Autoimmune hepatitis was diagnosed when clinical characteristics and available laboratory data indicated other causes of liver cirrhosis. Patients with both hepatitis and alcohol as the cause of cirrhosis were considered to have a viral etiology. The CP, MELD, and MELD-GRAIL-Na scores were calculated based on the clinical, laboratory, and radiologic results of the first day of admission. One- and three-month mortality were the primary outcomes of the study. Predictive values for 1- and 3-month mortality were compared between the CP, MELD, and MELD-GRAIL-Na scores.

### 2.1. Calculation of the CP, MELD, and MELD-GRAIL-Na Scores

The following formulas were used to calculate the scores [9,10].
MELD-GRAIL-Na = 29.751 + 10.836 * log(INR)+ 3.039 * log(bilirubin, mg/dL)− 5.054 * log(GRAIL, mL/min/1.73 m^2^)− 0.372 * log(Na, mEq/L)
where the lower and upper limits of bilirubin (1, infinite), international normalized ratio (INR) (1, 3), sodium (125, 140), and GRAIL (15, 90) are present. Dialysis was set as a GFR of 15 mL/min/1.73 m^2^. GRAIL was calculated using a website calculator provided by Baylor Scott & White Health, Dallas, TX 75246, USA. “www.bswhealth.med/grail (accessed on 15 March 2023)”.

### 2.2. Statistical Analysis

The CP score, MELD, and MELD-GRAIL-Na scores were calculated, and the predictive values for 1- and 3-month mortality were assessed using the area under the receiver operating curve (AUROC). *p*-values less than 0.05 were considered statistically significant. Comparison of AUROCs was performed using DeLong’s test through MedCalc version 12.7.2 (MedCalc Software, Mariakerke, Ghent, Belgium).

### 2.3. Ethics

This study was approved by the Institutional Review Board of Konkuk University Medical Center (Approval code: 2022-10-017).

## 3. Results

### 3.1. Baseline Characteristics

A total of 1249 patients were enrolled in the study. The baseline characteristics are shown in Table 1. Most patients were male (n = 842, 67.4%), and the mean age was 57.2 ± 12.9 years. The most common cirrhosis etiology was alcohol consumption (n = 765, 61.2%), followed by HBV (n = 210, 16.8%) and HCV (n = 70, 5.6%). Less common causes included autoimmune hepatitis (n = 37, 3.0%), primary biliary cirrhosis (n = 21, 1.7%), and NAFLD/NASH (n = 4, 0.3%). Average values were 4.3 ± 6.4 mg/dL for bilirubin, 3.1 ± 1.1 g/dL for albumin, 1.1 ± 0.9 mg/dL for creatinine, 22.2 ± 17.3 mg/dL for blood urea nitrogen, 135 ± 5.7 mEq/L for sodium, and 1.5 ± 0.7 for INR. The mean CP score was 8.1 ± 2.3, the mean MELD score was 15.2 ± 7.3, and the mean MELD-GRAIL-Na score was 22.3 ± 2.9. There were 387 patients classified as CP class A (31.0%), 505 patients as CP class B (40.4%), and 357 patients as CP class C (28.6%). CP classes B and C accounted for 69.0% of all enrolled patients (n = 862).

### 3.2. Clinical Course of Patients

The clinical course of patients who died within one and three months of initial admission is shown in Table 2. A total of 102 patients died within 1 month of admission, with a mortality rate of 8.2%. Of these patients, 37 (36.3%) died of hepatic failure, 21 (20.6%) died of variceal bleeding, and 14 (13.7%) died of diseases other than liver disease at the 3-month follow-up. Other causes of mortality included spontaneous bacterial peritonitis (7.8%), hepatorenal syndrome (6.9%), hepatic encephalopathy (4.9%), unknown causes (2.9%), and liver transplantation failure (2.0%). 

A total of 146 patients died within 3 months of admission, with a mortality rate of 11.7%. Among these patients, 56 (38.4%) died of hepatic failure, 24 (16.4%) of variceal bleeding, 17 (11.6%) of diseases other than liver disease, 11 (7.5%) of spontaneous bacterial peritonitis, 11 (7.5%) of unknown cause, 10 (6.8%) of hepatic encephalopathy, 7 (4.8%) of hepatorenal syndrome, and 4 (2.7%) of liver transplantation failure.

### 3.3. Baseline Characteristics of Patients with Advanced Cirrhosis (CP Classes B and C)

CP class A liver cirrhosis has a mild clinical status; hence, cirrhosis-related deaths are rare. In our study, only four patients with CP class A died within one month of admission. Of these patients, three died due to liver disease. To assess the relationship between cirrhosis and mortality, we performed an analysis of patients with advanced cirrhosis (CP classes B and C), excluding patients with CP class A. A total of 862 patients were enrolled (CP class B, 505; CP class C, 357). Baseline characteristics are shown in Table 3. Most patients were male (n = 583, 67.6%), with a mean age of 56.9 ± 13.2 years. The most common etiology of cirrhosis was alcohol consumption (n = 564, 65.4%), followed by HBV (n = 109, 12.6%) and HCV (n = 50, 5.8%). Less common causes included autoimmune hepatitis (n = 26, 3.0%) and primary biliary cirrhosis (n = 13, 1.5%). Average values were 5.8 ± 7.2 mg/dL for bilirubin, 2.9 ± 1.2 g/dL for albumin, 1.1 ± 1.0 mg/dL for creatinine, 23.3 ± 18.6 mg/dL for blood urea nitrogen, 134 ± 6.0 mEq/L for sodium, and 1.6 ± 0.8 for INR. The mean CP score was 9.2 ± 1.9, the mean MELD score was 17.7 ± 7.3, and the mean MELD-GRAIL-Na score was 23.3 ± 2.8, respectively.

### 3.4. Clinical Course of Patients with Advanced Cirrhosis (CP Classes B and C)

The causes of death and number of deaths with advanced cirrhosis are shown in Table 4. No significant changes were observed in the cause of death between patients with CP classes B and C. The most common causes of bleeding were hepatic and varix bleeding.

### 3.5. AUROC Values and Score Comparison

#### 3.5.1. One-Month Mortality

The AUROCs of the CP score, MELD, and MELD-GRAIL-Na for 1-month mortality are shown in Figure 1a–c, respectively. The AUROC of the CP score was 0.831 (*p* < 0.0001), that of MELD was 0.847 (*p* < 0.0001), and that of MELD-GRAIL-Na was 0.857 (*p* < 0.0001). Figure 1d shows a comparison of these AUROCs. There were no significant differences in the 1-month mortality prediction between the CP, MELD, and MELD-GRAIL-Na scores.

#### 3.5.2. Three-Month Mortality

The AUROCs of the CP, MELD, and MELD-GRAIL-Na scores for 3-month mortality are shown in Figure 2a–c, respectively. The AUROC of the CP score was 0.837 (*p* < 0.0001), that of MELD was 0.827 (*p* < 0.0001), and that of MELD-GRAIL-Na was 0.835 (*p* < 0.0001). A comparison of the AUROCs is shown in Figure 2d. There were no significant differences in the 3-month mortality prediction between the CP, MELD, and MELD-GRAIL-Na scores.

#### 3.5.3. One-Month Mortality Prediction in Patients with CP Classes B and C

The AUROCs of the CP, MELD, and MELD-GRAIL-Na scores for 1-month mortality in patients with CP classes B and C are shown in Figure 3a–c, respectively. The AUROC of the CP score was 0.782 (*p* < 0.0001), that of MELD was 0.809 (*p* < 0.0001), and that of MELD-GRAIL-Na was 0.825 (*p* < 0.0001). A comparison of the AUROCs is shown in Figure 3d. There was a significant difference between the CP score and the MELD-GRAIL-Na (*p* = 0.0428), MELD, and MELD-GRAIL-Na scores (*p* = 0.0493). Bayesian positive predictive value (PPV) and negative predictive value (NPV) of model/scores in predicting mortality were PPV 30.42, NPV 95.04 for CP score, PPV 30.34, NPV 95.7 for MELD and PPV 31.06, NPV 96.01 for MELD-GRAIL-Na. MELD-GRAIL-Na was high for both PPV and NPV.

#### 3.5.4. Three-Month Mortality Prediction in Patients with CP Classes B and C

The AUROCs of the CP, MELD, and MELD-GRAIL-Na scores for 3-month mortality are shown in Figure 4a–c, respectively. The AUROCs of the CP, MELD, and MELD-GRAIL-Na scores were 0.775 (*p* < 0.0001), 0.769 (*p* < 0.0001), and 0.786 (*p* < 0.0001), respectively. A comparison of the AUROCs is shown in Figure 4d. Regarding the 3-month mortality prediction in patients with CP classes B and C, there were significant differences between the MELD and MELD-GRAIL-Na scores (*p* = 0.0225), but no significant difference was found between the CP score and MELD-GRAIL-Na. Bayesian positive predictive value (PPV) and negative predictive value (NPV) of model/scores in predicting mortality were PPV 39.63, NPV 91.32 for CP score, PPV 35.29, NPV 92.2 for MELD and PPV 33.54, NPV 94.16 for MELD-GRAIL-Na. MELD-GRAIL-Na was high in NPV, but low for PPV.

## 4. Discussion

Over the years, ongoing efforts have been made to estimate and predict mortality in patients with cirrhosis. The CP scoring system could be easily used as it comprised five variables (i.e., serum bilirubin, albumin, prothrombin time, ascites, and encephalopathy). Since the presence and amount of ascites and grade of encephalopathy are a subjective variable, efforts to overcome subjectivity may have led to the development of MELD. MELD has been the optimal scoring system for mortality prediction and transplant allocation prioritization, but as new studies and research have been introduced, MELD has demonstrated some shortcomings. Studies have shown that hyponatremia and renal function are crucial factors for morbidity and mortality among patients with cirrhosis [4,5,10,11]. MELD-Na, Refit-MELD, and Refit-MELD-Na are a result of the integration of sodium and renal function into the MELD [2,3,4]. Recently, the newly designed ‘GRAIL’ model has been introduced to better estimate GFR in patients with liver diseases, as it more heavily weighs the role of renal function [9]. MELD-GRAIL-Na, the integration of GRAIL with MELD-Na, was introduced in 2020, suggesting its potential superiority compared to MELD and MELD-Na [10].

This study aimed to review the efficacy of MELD-GRAIL-Na in Korean patients with cirrhosis. In the original study, 4.1% of the study cohort were Asian, suggesting that the Asian race may have not been adequately represented. Although this was a retrospective study, the large population may represent the efficacy of MELD-GRAIL-Na in predicting short-term mortality in Koreans. Our study population had baseline characteristics similar to those of the original article. The following parameters were compared: age (57 vs. 57.2 years), female (37.8% vs. 32.6%), diabetes (28.0% vs. 27.9%), serum creatinine (1.0 vs. 1.1), INR (1.42 vs. 1.5), albumin (3.1 vs. 3.1), sodium (137 vs. 135), and MELD (17 vs 15.2). The similarity in the values of these variables may have contributed to the reliability of the comparisons in this study. Major differences were noted in serum bilirubin (2.8 vs. 4.3), percentage of dialysis patients (9.3% vs.0.01%), and cirrhosis etiology. The etiology of cirrhosis in the original article was alcohol (26.7%), cryptogenic/NASH (20%), HCV (18.3%), and others (35%), while our study included alcohol (61.2%), HBV (16.8%), HCV (5.6%), and cryptogenic/NASH (1.5%). These similarities in baseline characteristics, especially the percentage of females and baseline MELD scores, may explain the reliability of our study in the Korean population.

MELD-GRAIL-Na was superior to the CP and MELD scores in the prediction of short-term mortality, but statistically significant differences were found only in patients with advanced cirrhosis (CP classes B and C). This result is thought to be due to the fundamentals of GRAIL and the enrolled patients with CP class A. The fundamentals of GRAIL were to improve the classification of patients with low GFR, and it has been shown to classify a GFR lower than 40 mL/min/1.72 m^2^ better than alternative models, such as CKD-EPI, MDRD-4, and MDRD-6. However, classifications at high GFR were lower with GRAIL than other alternative equations. For values of GFR greater than 45 mL/min/1.72 m^2^, the correct classification through the GRAIL, CKD-EPI, MDRD-4, and MDRD-6 scores were 89.3%, 96.3%, 93.2%, and 95.2%, respectively [9]. Patients with CP class A have fewer cirrhosis-related complications and generally have higher GFR than patients with advanced cirrhosis (CP classes B and C). Our study enrolled 387 patients with CP class A (31.0%), and the average GRAIL and creatinine levels for CP class A patients were 98.4 mL/min/1.73 m^2^, while those for CP classes B and C were 84.0 mL/min/1.73 m^2^. A large percentage of patients with CP class A and the fundamentals of GRAIL may explain the statistical insignificance of MELD-GRAIL-Na over the CP and MELD scores for all enrolled patients. 

Our study has some limitations. First, our hospital had very few liver transplant cases; therefore, various patients with mild to severe statuses were included in the study rather than wait-list patients only. Patients admitted for simple symptom control and healthcare checkups were included. Second, very few patients underwent dialysis. The effects of dialysis on GRAIL levels are not shown. Third, the dependence of alcohol as major etiology of cirrhosis. Cirrhosis patients with an alcohol-based etiology comprised more than half of the total sample in our study (61.2%). While the incidence of alcoholic cirrhosis is increasing in Korea, the major cause of cirrhosis in Korea is hepatitis B virus [12]. Vaccination and use of antiviral medications have resulted in better outcomes in cirrhosis patients with HBV [13,14], and many patients are under outpatient follow-up. Our study enrolled admitted patients, which may have contributed to the lack of cirrhosis with HBV-based etiology in our study. Moreover, it is known that cirrhosis progression and outcomes are worse in patients with alcoholic cirrhosis than in those with other etiologies [15]. Our biased proportion of cirrhosis patients with an alcohol-based etiology could have affected the mortality rate predictions in this study.

## 5. Conclusions

Compared to CP and MELD, MELD-GRAIL-Na was found to be a better and more useful system for evaluating short-term (1- and 3-month) mortality in Korean patients with cirrhosis, especially those with advanced cirrhosis (CP classes B and C).

## Figures and Tables

**Figure 1 medicina-59-00592-f001:**
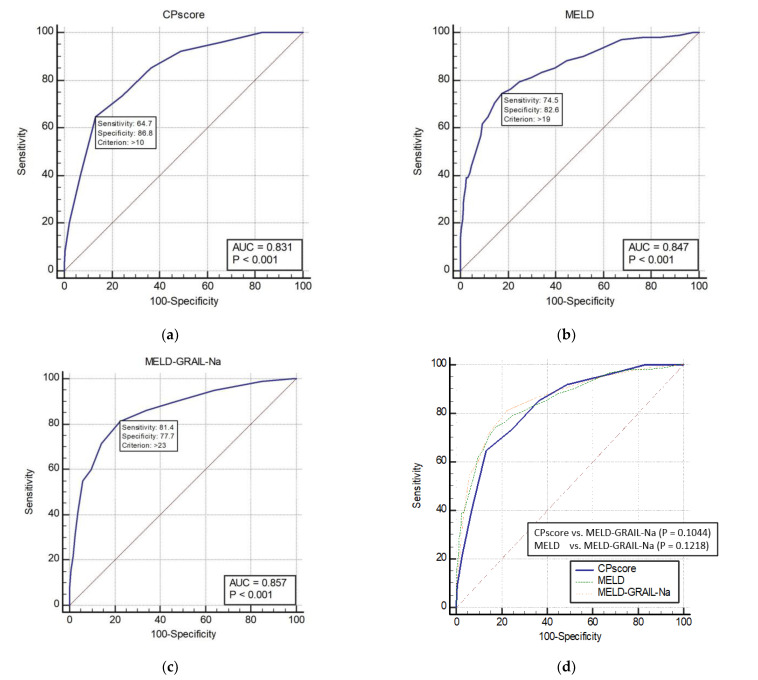
Receiver operating characteristic curves (ROC) and area under the ROC (AUROC) of patients with total cirrhosis for 1-month mortality. (**a**) CP score for 1-month mortality, 0.831 (95% confidence interval [CI], 0.809–0.851); (**b**) MELD for 1-month mortality, 0.847 (95% CI, 0.826–0.867); (**c**) MELD-GRAIL-Na for 1-month mortality, 0.857 (95% CI, 0.836–0.876); (**d**) comparison of AUROC for 1-month mortality between the CP, MELD, and MELD-GRAIL-Na scores.

**Figure 2 medicina-59-00592-f002:**
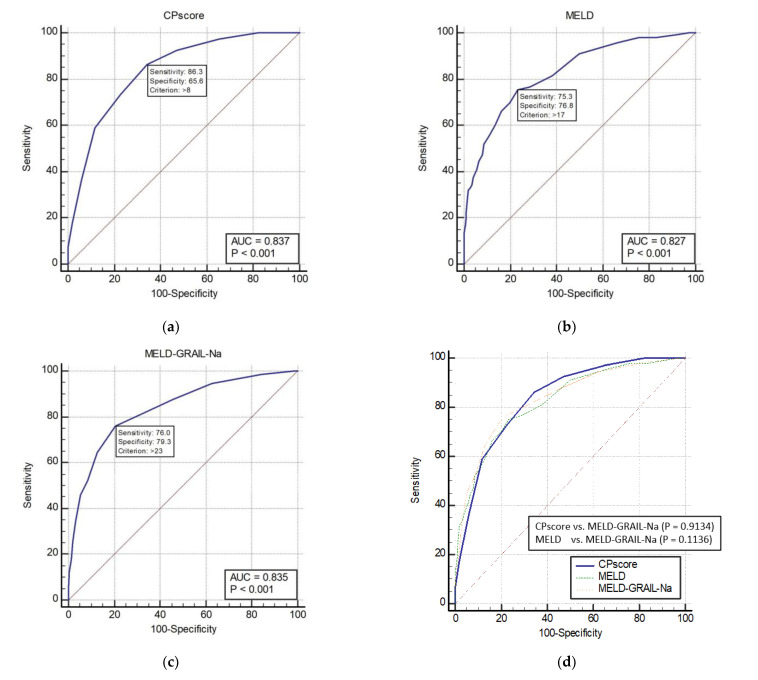
Receiver operating characteristic curves (ROC) and area under the ROC (AUROC) of patients with total cirrhosis for 3-month mortality. (**a**) CP score for 3-month mortality, 0.837 (95% CI, 0.815–0.857); (**b**) MELD for 3-month mortality, 0.827 (95% CI, 0.805–0.848); (**c**) MELD-GRAIL-Na for 3-month mortality, 0.835 (95% CI, 0.814–0.856); (**d**) comparison of AUROC for 3-month mortality between the CP, MELD, and MELD-GRAIL-Na scores.

**Figure 3 medicina-59-00592-f003:**
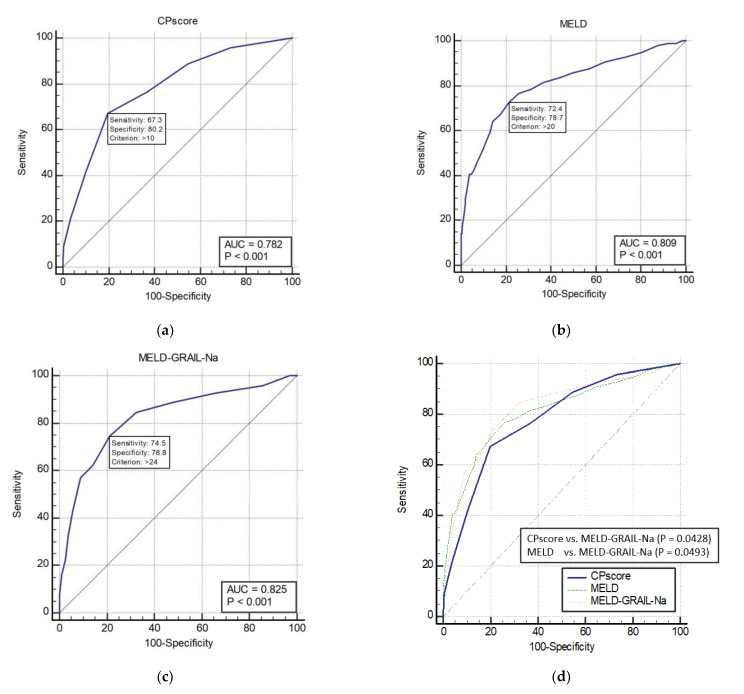
Receiver operating characteristic curves (ROC) and area under the ROC (AUROC) of patients with CP classes B and C for 1-month mortality. (**a**) CP score for 1-month mortality, 0.782 (95% CI, 0.753–0.809), (**b**) MELD for 1-month mortality, 0.809 (95% CI, 0.781–0.834); (**c**) MELD-GRAIL-Na for 1-month mortality, 0.825 (95% CI, 0.798–0.850); (**d**) comparison of AUROC for 1-month mortality between the CP, MELD, and MELD-GRAIL-Na scores.

**Figure 4 medicina-59-00592-f004:**
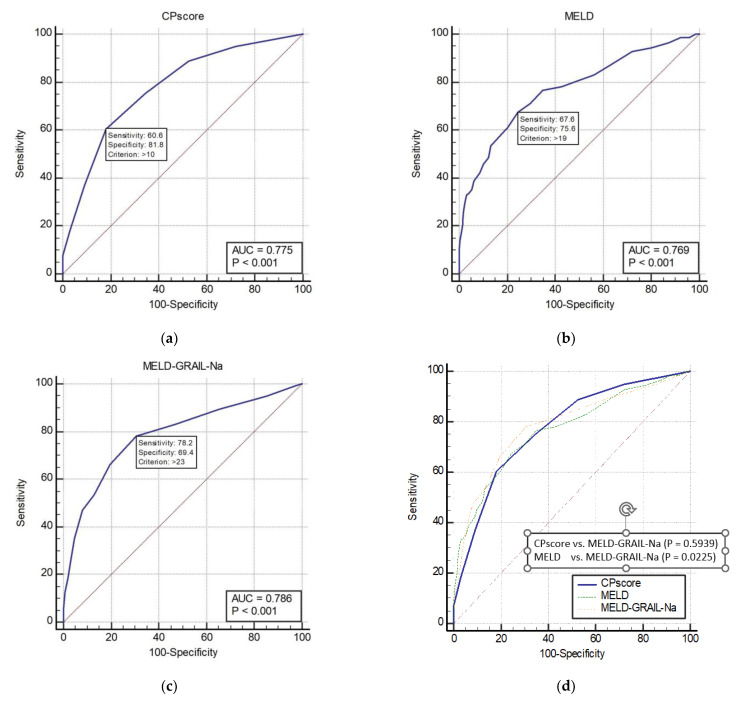
Receiver operating characteristic curves (ROC) and area under the ROC (AUROC) of patients with CP classes B and C for 3-month mortality. (**a**) CP score for 3-month mortality, 0.775 (95% CI, 0.746–0.803); (**b**) MELD for 3-month mortality, 0.769 (95% CI, 0.739–0.797); (**c**) MELD-GRAIL-Na for 3-month mortality, 0.786 (95% CI, 0.757–0.813); (**d**) comparison of AUROC for 3-month mortality between the CP, MELD, and MELD-GRAIL-Na scores.

**Table 1 medicina-59-00592-t001:** Baseline characteristics of the enrolled patients.

Variables	Number of Subjects (Alcohol/HBV/HCV)
Total patients	1249 (765/210/70)
Gender, male	842 (608/40/30)
Age (years)	57.2 ± 12.9 (53.6/57.0/62.4)
Serum bilirubin (mg/dL)	4.3 ± 6.4
Serum albumin (g/dL)	3.1 ± 1.1
Serum creatinine (mg/dL)	1.1 ± 0.9
Serum blood urea nitrogen (mg/dL)	22.2 ± 17.3
Serum sodium (mEq/L)	135 ± 5.7
Prothrombin tine (INR)	1.5 ± 0.7
CP class A	387 (201/101/20)
CP class B	505 (304/64/35)
CP class C	357 (260/45/15)
CP score	8.1 ± 2.3 (8.4 ± 2.4/7.3 ± 2.2/7.9 ± 2.1)
MELD score	15.2 ± 7.3 (16.4 ± 7.8/13.0 ± 6.4/14.2 ± 6.2)
MELD-GRAIL-Na score	22.3 ± 2.9 (22.7 ± 3.1/21.4 ± 2.5/22.0 ± 2.5)

Continuous variables are presented as mean ± standard deviation. Abbreviations: HBV, hepatitis B virus; HCV, hepatitis C virus; INR, international normalized ratio; CP, Child–Pugh; MELD, Model for End-Stage Liver Disease; and MELD-GRAIL-Na, Model for End-stage Liver Disease–Glomerular Filtration Rate Assessment in Liver Disease–Sodium.

**Table 2 medicina-59-00592-t002:** One- and three-month mortality and causes of death.

Variables	Number of Subjects (Alcohol/HBV/HCV)	
	1-Month	3-Month
Mortality	102 (71/12/2)	146 (99/13/5)
**Cause of death**		
Varix bleeding	21 (17/3/0)	24 (20/3/0)
Hepatic encephalopathy	5 (2/0/1)	10 (6/0/1)
Spontaneous bacterial peritonitis	8 (7/0/0)	11 (10/0/0)
Hepatorenal syndrome	7 (3/2/0)	7 (3/2/0)
Hepatic failure	37 (27/4/1)	56 (37/5/3)
Other than liver disease	14 (8/1/0)	17 (9/1/0)
Liver transplantation failure	2 (2/0/0)	4 (4/0/0)
Unknown	8 (5/2/0)	17 (10/2/0)

Abbreviations: HBV, hepatitis B virus; HCV, hepatitis C virus.

**Table 3 medicina-59-00592-t003:** Baseline characteristics of enrolled patients (CP classes B and C).

Variables	Number of Subjects (Alcohol/HBV/HCV)
Total patients	862 (564/109/50)
Gender, male	583 (448/74/29)
Age (years)	56.9 ± 13.2 (52.7/56.9/62.5)
Serum bilirubin (mg/dL)	5.8 ± 7.2
Serum albumin (g/dL)	2.9 ± 1.2
Serum creatinine (mg/dL)	1.1 ± 1.0
Serum blood urea nitrogen (mg/dL)	23.3 ± 18.6
Serum sodium (mEq/L)	134 ± 6.0
Prothrombin tine (INR)	1.6 ± 0.8
CP class B	505 (304/64/35)
CP class C	357 (260/45/15)
CP score	9.2 ± 1.9 (9.4 ± 1.9/9.1 ± 1.7/8.8 ± 1.7)
MELD score	17.7 ± 7.3 (18.8 ± 7.6/16.5 ± 6.9/16.2 ± 6.2)
MELD-GRAIL-Na score	23.3 ± 2.8 (23.6 ± 2.9/22.7 ± 2.7/22.8 ± 2.4)

Continuous variables are presented as mean ± standard deviation. Abbreviations: HBV, hepatitis B virus; HCV, hepatitis C virus; INR, international normalized ratio; CP, Child–Pugh; MELD, Model for End-Stage Liver Disease; MELD-GRAIL-Na, Model for End-stage Liver Disease–Glomerular Filtration Rate Assessment in Liver Disease–Sodium.

**Table 4 medicina-59-00592-t004:** One- and three-month mortality and causes of death (CP classes B and C).

Variables	Number of Subjects (Alcohol/HBV/HCV)	
	1-Month	3-Month
Mortality	98 (70/11/1)	142 (71/12/2)
**Cause of death**		
Varix bleeding	21 (17/3/0)	24 (21/2/0)
Hepatic encephalopathy	4 (2/0/0)	9 (6/0/0)
Spontaneous bacterial peritonitis	8 (7/0/0)	11 (10/0/0)
Hepatorenal syndrome	7 (3/2/0)	7 (3/2/0)
Hepatic failure	37 (27/4/1)	56 (37/5/3)
Other than liver disease	11 (7/0/0)	14 (8/1/0)
Liver transplantation failure	2 (2/0/0)	4 (4/0/0)
Unknown	8 (5/2/0)	17 (10/2/1)

Abbreviations: HBV, hepatitis B virus; HCV, hepatitis C virus.

## Data Availability

Not applicable.

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
