# Peer review of "MELD-GRAIL-Na Is a Better Predictor of Mortality Than MELD in Korean Patients with Cirrhosis"

_medicina, 2023, doi:10.3390/medicina59030592_

Round 1
Reviewer 1 Report (Previous Reviewer 1)
I congratulate the authors for this improved work. I believe most of my comments have been addressed.
Reviewer 2 Report (Previous Reviewer 2)
It seems to be ok for publication.
Reviewer 3 Report (Previous Reviewer 3)
None
This manuscript is a resubmission of an earlier submission. The following is a list of the peer review reports and author responses from that submission.
Round 1
Reviewer 1 Report
Overall, a well-designed and presented work validating the prognostic value of MELD-GRAIL-Na in a population of Korean. Authors show that MELD-GRAIL-Na can fairly predict 1- and 3-month mortality in patients with Child Pugh C and B cirrhosis. My comments and suggestions are as follows.
MAJOR
1. Considering that MELD-GRAIL-Na showed just 1% improvement over MELD for such a significant sample size (1249 patients), the “superiority” is not so superior. I recommend the authors to tone down the conclusion statement to reflect the findings.
2. The discussion generally needs a complete overhaul as the argument for the use of MELD-GRAIL-Na and the context in terms of current prognostic markers including the ACLF (acute on chronic liver disease ) score has not been mentioned. Some other specifics are mentioned below.
3. For clinical applicability, it would interesting if authors can calculate the Bayesian positive (PPV) and negative predictive value (NPV) of the MELD-GRAIL-Na considering the mortality rate within the studied population. Perhaps this could make a stronger case for the predictive value of this model/score.
MINOR
I. Line 206-209: the statement about CP score needs references
II. Result section: where data is already presented in the tables, summarize the data rather than repeating the table contents e.g., Line 119-121, the other causes of death statement is not necessary since this data is already in the table 2.
III. Line 202-203: “…differences between the MELD…”.
IV. Line 209: “…However, variables such as ascites and encephalopathy may result in unreliable prognostic valued due to their subjectivity…” Please rephrase as such and provide relevant references.
V. Line 256-257: “…Third, the dominant….study” This statement is not clear please clarify by rephrasing.
Reviewer 2 Report
The subject of prognosticating cirrhotic patients is a very relevant one. I have liked this paper, it is well written, and overal interesting. Statistics is correct. I just suggest the authors compare these scores with CLIF-SOFA and CLIF-C scores, which have been constantly superior in prognosticating these patients, as pointed out by a recently published systematic review: https://www.wjgnet.com/1948-5182/full/v14/i12/2025.htm . Besides these, some additional references on this topic:
https://www.scielo.br/j/ag/a/gN3s94VdKdHLLkcKF9SGwwH/abstract/?lang=en
Reviewer 3 Report
The authors of this is article compare the prognostic value of the Child-Pugh (CP), MELD-Na and MELD-GRAIL-Na scores in predicting survival for 1 and 3 months of cirrhotic patients. Their cohort comprises of 1245 patients with cirrhosis of various etiology, followed at a University Korean Hospital for a mean time of 5 years. The article shows that the predictive value of MELD-GRAIL-Na score is significantly better than CP and MELD-Na score in patients with advanced cirrhosis (CP score B and C), but not in patients with CP score A. The question whether the MELD-GRAIL-Na score is superior to the MELD-Na has not been settled as yet and this article adds another piece to controversy. It is also apparent that, that MELD is not perfect and this article represents one of the many efforts to improve it.
The study is well written, concise and executed with satisfactory proficiency. Despite the fact that it includes only Asian (Korean) patients and the alcoholic etiology weighs much more than viral, its results coincide with another study with non-Asian patients with advanced cirrhosis of viral etiology (Hepatology, 2019, 32:1343).